# Should I Buy the Current Narrative about Predatory Journals? Facts and Insights from the Brazilian Scenario

**Cicero Cena** [1,*] **, Daniel A. Gonçalves** [2] **and Giuseppe A. Câmara** [3]

1 SISFOTON-Laboratório de Óptica e Fotônica, UFMS-Universidade Federal de Mato Grosso do Sul, Campo Grande 79070-900, MS, Brazil
2 Faculty of Exact Science and Technology, UFGD-Universidade Federal da Grande Dourados, Dourados 79804-970, MS, Brazil; daniel.araujogoncalves@gmail.com
3 LEMA, Institute of Chemistry, UFMS-Universidade Federal de Mato Grosso do Sul, Campo Grande 79070-900, MS, Brazil; giuseppe.silva@ufms.br
* Correspondence: cicero.cena@ufms.br

**Abstract:** The burgeoning landscape of scientific communication, marked by an explosive surge in published articles, journals, and specialized publishers, prompts a critical examination of prevailing assumptions. This article advocates a dispassionate and meticulous analysis to avoid policy decisions grounded in anecdotal evidence or superficial arguments. The discourse surrounding so-called predatory journals has been a focal point within the academic community, with concerns ranging from alleged lack of peer review rigor to exorbitant publication fees. While the consensus often leans towards avoiding such journals, this article challenges the prevailing narrative. It calls for a more nuanced understanding of what constitutes predatory practices and underscores the importance of skeptical inquiry within our daily academic activities. The authors aim to dispel misconceptions and foster a more informed dialogue by scrutinizing APCs, impact factors, and retractions. Furthermore, the authors delve into the evolving landscape of scientific publishing, addressing the generational shifts and emerging trends that challenge traditional notions of prestige and impact. In conclusion, this article serves as a call to action for the scientific community to engage in a comprehensive and nuanced debate on the complex issues surrounding scientific publishing.

**Keywords:** predatory journals; scientometric; bias





The burgeoning landscape of scientific communication, marked by a booming growth in published articles, journals, and specialized publishers, prompts a critical examination of prevailing assumptions. This article advocates for a dispassionate and meticulous analysis to avoid policy decisions grounded in anecdotal evidence or superficial arguments.

The discourse surrounding so-called predatory journals has been a focal point within the academic community since the Beall and Cabells list [1,2]. The main concerns include exorbitant publication fees, low quality of work, journal titles reminiscent of prestigious journals, and quick acceptance, which raises doubts about the reliability of the peer-review process adopted by these journals.

By accepting that these assumptions are true, it seems easy to argue that such "predatory journals" should be avoided, as no well-intentioned scientists would willingly expose themselves to this kind of "abusive relationship". In other words, after years of hard work where an idea is transformed into scientific data by the small "miracle" of human inspiration, it is regrettable to discover that the chosen publisher denigrates the quality of your work because the scientific community views it with suspicion (or even disbelief) since you published it in an unreliable platform. The problem is not exclusive to Brazil, but it is worsened when the country does not have a consolidated national policy that allocates specific resources for the payment of open access publication fees.

Therefore, instead of simply accepting that the expansion in the number of journals occurred primarily via a proliferation of suspect journals, we invite the community to delve more deeply into who defines and how we define what predatory journals are. To do so, it is necessary to employ the skepticism that is so cherished in our daily academic activities and explore the topic, its potential consequences, and even the underlying interests that may influence decision-makers.

In Brazil, master's and PhD programs are evaluated every four years by an agency directly subordinate to the Ministry of Education (CAPES). Throughout this interval, the heads of post-graduation programs meet at CAPES headquarters to deliberate on the key parameters chosen for comparative analysis during the evaluation process. Recently, after one of those meetings, a non-official, highly threatening message circulated on WhatsApp, derived from a post on the "Predatory Journals" website that placed the publishers MDPI, Hindawi, and Frontiers "behind bars" [3]. It seems that behind the stage of science, a handful of people (whose interests we ignore) want to decide which scenario will be used for the scene without talking with those who are writing the script.

Apparently, part of the Brazilian scientific community with decision-making power has chosen to accept, as an act of faith, the post on the mentioned website. They simply assumed MDPI, Hindawi, and Frontiers to be predatory without distinguishing between the journals published by these entities. However, the prestige of a journal is primarily defined by the quality of the editorial board and its ability to appoint reviewers capable of (a) discerning, among the proposed publications, which ones are relevant and (b) foreseeing, to some extent, which research can bring visibility to the journal. When these criteria are applied to the peer-review system, with all the inherent idiosyncrasies of the individuals who make up the system, they generate a significant heterogeneity of prestige among journals, even those published under the same editorial label.

Returning to the post, the suggestive image places the logos of these publishers behind bars with the subtitle "major predatory publishers". The post mentions that starting from January 2023, Zhejiang Gongshang University would no longer consider articles published by these publishers in its academic performance evaluations. The text also suggests that the "banning" of these publishers by CAPES is an ongoing process. Yet, the post ends arguing that we should follow the example of Zhejiang Gongshang University, one of the top three Chinese universities (it is worth noting that according to Times Higher Education, the mentioned university is not among the top three in its country). The original post also appeals to a certain "trickery" of the training researchers, warning them that their performance depends on the opinion of their employers and that "wasting money to get your papers accepted in the shortest possible time may not help you in the long term" [3].

Well, some skepticism is welcome when science administrators in Brazil spread unofficial information with the intention of influencing the community to advocate for interests that remain undisclosed. This echoing of "truths" is mere rhetoric and must be scrutinized, as will be demonstrated here. One hypothesis is that the bearer of the message himself did not critically analyze it before accepting it as truth. This assumption is especially troubling when considering its potential position in the decision-making hierarchy of science and technology in our country, subjecting our entire community to rules guided by a biased viewpoint.

To substantiate the discussion, let us analyze some data from two major publishers, Elsevier and MDPI. The first one is a well-established publisher with numerous prestigious journals across various fields of knowledge, while MDPI is an alleged "predatory publisher". The Article Processing Charges (hereafter referred to as APC) of MDPI, which encompasses 422 journals, range from 500 to 2900 Swiss Francs, with an average value of approximately 1500 Swiss Francs (around USD 1670). On the other hand, Elsevier has 2729 titles [4], with APCs ranging from USD 200 to 10,400 and an average APC of USD ~2900.

Early critics may argue that they do not pay to publish in Elsevier, but they overlook the fact that CAPES has two access contracts with Elsevier that together add up to around

USD 127 million [5]. Furthermore, there is a collective assumption that Elsevier values the quality of its publications, unlike MDPI, which would be a factor in classifying the latter as predatory. However, although we do not have the statistics, we and many of our colleagues review manuscripts for both platforms. Therefore, if the reviewers define the quality of a journal, and if the pool of reviewers is similar in both cases, what justifies the discrepancy in the quality of works between the analyzed publishers? Furthermore, if the papers published by MDPI are of low quality, why do their journals have increasingly higher impact factors? Does the scientific community tend to cite poor-quality works as an example? What kind of science are we practicing then?

Continuing this reasoning, we extracted the number of papers published from 2019 to 2023 for both publishers using the Web of Science platform. In this period, Elsevier and MDPI published 3,167,221 and 1,172,615 papers, respectively. Cross-referencing these numbers with the retracted papers by the same publishers, through a query in the Retraction Watch Database, we observed that Elsevier and MDPI had 2305 (0.07%) and 142 (0.01%) retracted papers, respectively, during the period. Also, the median impact factor of 528 Elsevier journals classified as devoted to the Chemistry field in the Qualis 2017–2020 (a scale of prestige used by CAPES) is 4.45, while for MDPI's 66 journals, it is 3.70. This difference should not be overlooked, but we cannot be absolutists. Similar to a researcher's H-index increasing with the number of publications and over time, it is expected that something analogous would happen with the impact factor of MDPI journals. A complete understanding of this scenario would require a statistical analysis beyond the scope of this text.

A key aspect that challenges claims without evidence is the scrutiny of published works, a characteristic supposedly absent in predatory journals. In MDPI, many reviews are available alongside the publication, allowing for an evaluation of the entire review process until the manuscript is accepted. Furthermore, in some cases, it is even possible to identify the reviewers and assess whether they are experts in the field.

Another counterpoint is the three-year contract to access the Elsevier database (2023–2025). We cannot predict the future, but from 2021 to 2023, Brazilian scientists published 52,299 articles in Elsevier. If we had paid the average APC (USD ~2900) in the open access model, the investment would have been approximately 1.2 times the current contract. It is important to highlight that this contract expires on 31 December 2025, while the timeframe for open access is indefinite. If everyone is in a similar situation, maybe we should all pay APCs, and become free from the paywall model.

The data presented here provide enough material to outline the present scenario. In essence, the available data do not justify the banishment of certain publishers from the scientific scene, even though there may have been lapses in the practice of individual journals. The underlying suggestion is that we should be more cautious before adopting a binary logic. Furthermore, instances of misconduct practices are also observed in supposed "serious publishers". Let us not forget recent examples from PLOS ONE and Scientific Reports [6].

We believe the prejudice against new publishers is partially based on the following scenario: the management of science in Brazil lies in the hands of senior researchers who were trained in a different scientific environment. Most of them have well-established groups with a good infrastructure and abundant workforce. Hence, it is easier for these groups to maintain a high rate of scientific production (10+ papers per year). Since the volume of submissions is high, the flow of publications remains at the same level, even if it requires waiting for the longer processing time (including resubmissions) of publishers as Elsevier.

On the other hand, the relatively shorter times of processing submissions of publishers like MDPI allow researchers from smaller groups, typically younger, to give momentum to their intermittent production, eventually reaching similar numbers of papers.

This scenario suggests a generational friction point in the Brazilian scientific community: Thanks to the rise of short-time processing publishers, the scientific production rate

in smaller centers may be approaching that of the top Brazilian universities. Let us test this hypothesis: in a query regarding the Scopus database, we compared the number of publications weighted by the number of authors from 2000 to 2023 between a prestigious Brazilian university and an emerging one (Figure 1). For this, we chose three of the top ten Brazilian universities according to the World University Ranking 2023 [7] to represent the consolidated ones. The top three universities in the Central-West region are the emerging ones chosen for comparison. It is noted that from 2015, there is a more pronounced inflection in the curve of emerging universities, partly explained by the renewal of the teaching staff, thanks to programs that promoted the expansion and universalization of postgraduate studies in Brazilian public universities [8]. However, there is a more pronounced growth for one of the emerging institutions, coinciding with the adoption of an internal subsidy policy for APCs starting in 2017. It is possible that this advent partially justifies the behavior expressed in Figure 1.

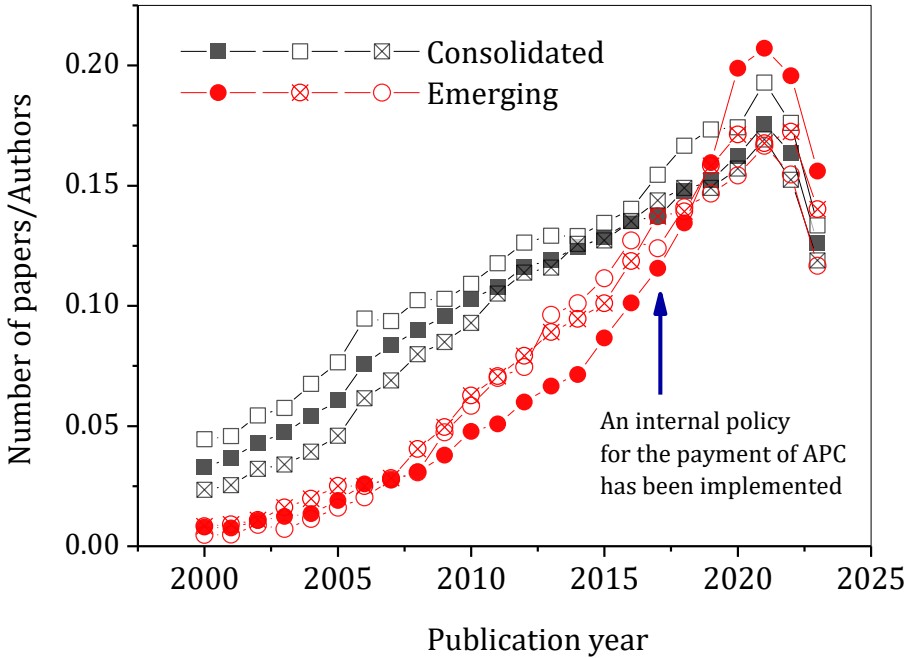

**Figure 1.** Published papers between 2000 and 2022, according to SCOPUS, weighted by the number of authors affiliated with examples from consolidated and emerging universities selected for comparison.

One last aspect deserves a brief comment: publishers like MDPI enable young researchers to be guest editors for special volumes. This policy minimizes the comparative advantages that well-established graduate programs had until recently in the evaluation cycles promoted by CAPES.

A colleague once reflected that "The dominant force will always resist the emerging one. . . even though both have problems, the more powerful ones can shape the narrative that suits them. . . but that's the game. . . and the emerging publishers have to respond to this debate."

This essay does not aim to exhaust the subject, and we acknowledge that there is still much to be considered on the topic. Our reflections are primarily a call to action for the scientific community to engage in a comprehensive and nuanced debate on the complex issues surrounding scientific publishing. Certainly, we can devise smarter solutions for the current model, but we will not do so without a broad debate, where the arguments of various stakeholders are considered. It is time to debate policy decisions based on evidence, considering the diverse interests and perspectives within the scientific ecosystem, rather than in "hallway conversation".

**Author Contributions:** Conceptualization: C.C. Methodology: C.C., D.A.G. and G.A.C. Investigation: C.C. and D.A.G. Visualization: C.C. and G.A.C. Writing—original draft: C.C. Writing—review and editing: C.C. and G.A.C. All authors have read and agreed to the published version of the manuscript.

**Funding:** This research received no external funding.

**Data Availability Statement:** The data presented in this study are available on request from the corresponding author.

**Acknowledgments:** CNPq–Conselho Nacional de Desenvolvimento Científico e Tecnológico, 302525/2022-0, 309810/2019-1.

**Conflicts of Interest:** The authors declare that they have no competing interests.

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
