# Peer review of "Should I Buy the Current Narrative about Predatory Journals? Facts and Insights from the Brazilian Scenario"

_publications, doi:10.3390/publications12010007_

Round 1
Reviewer 1 Report
Comments and Suggestions for Authors
The manuscript is a timely and original opinion paper on a topic that is relevant not only in Brazil but also in other countries, like in Germany or France where research performing organizations started to communicate in a similar way. I read it with interest and pleasure.
As an opinion paper, you don't need many references. However, in order to increase the readability and the interest of your paper for an international audience, I would suggest three improvements:
(1) Even if the paper is rather shoryt, I would suggest 4 or 5 sections with different headings (e.g., line 13 the issue, line 38 the case of Brazil, line 77 Frontiers and Elsevier, line 120 a generational friction point, line 161 a need for nuanced debate).
(2) At the beginning, add perhaps one or two references to the "discourse surrounding etc." (line 18) and to the "assumptions" (line 23). You don't mention Beall, and you don't mention Cabells; however, both lists are at the origin of this affair.
(3) At the end (or somewhere else, when speaking of Brazil), perhaps you should tell the audience that there are similar approaches and decisions in other countries and research organizations.
These are suggestions, not requirements :)
Author Response
- Even if the paper is rather shoryt, I would suggest 4 or 5 sections with different headings (e.g., line 13 the issue, line 38 the case of Brazil, line 77 Frontiers and Elsevier, line 120 a generational friction point, line 161 a need for nuanced debate).
Thank you for the suggestion, but the paper was designed as an opinion manuscript, which does not apply sections. We believe that in the current format, the readability of the text is better achieved.
(2) At the beginning, add perhaps one or two references to the "discourse surrounding etc." (line 18) and to the "assumptions" (line 23). You don't mention Beall, and you don't mention Cabells; however, both lists are at the origin of this affair.
Thank you for the suggestion; we added a mention in the text about the Beall and Cabell list.
(3) At the end (or somewhere else, when speaking of Brazil), perhaps you should tell the audience that there are similar approaches and decisions in other countries and research organizations.
Thank you for your suggestion; we added this information in the new version of the opinion manuscript.
Reviewer 2 Report
Comments and Suggestions for Authors
First of all, I think it is valuable to discuss the impact of perceived quality on the career of academics. However, in my opinion there are several issues with the manuscript that need to be addressed.
Title of the manuscript and the content
The title suggests a global and thorough analysis. The actual manuscript discusses the relative merits of two commercial publishers of open access articles in a Brazilian context. In my opinion, these do not match and I would strongly suggest to review the title.
Context: open access
The manuscript discusses the assessment of open access articles by comparing Elsevier and MDPI. This seems to suggest that APC based publishing of articles is the only way to publish in open access. Given the role of SciELO in Brazil, I would at least expect a short explanation why the discussion only focuses on the two publishers.
Context: assessment
There is an extensive discussion on the merit (or lack thereof) of the Journal Impact Factor and assessment of academic s in general. While a complete discussion of this subject lies beyond the scope of this manuscript - the discussion would probably barely fit in a book - some mention of this discussion would help to put the discussed subject in context.
Conclusion
In my view, the main argument of the manuscript centres around a generational difference in opinion on the value of articles published in Elsevier vs. MDPI in a Brazilian context. I would suggest to make this more clear, both in the title and the abstract.
Author Response
The title suggests a global and thorough analysis. The actual manuscript discusses the relative merits of two commercial publishers of open access articles in a Brazilian context. In my opinion, these do not match and I would strongly suggest to review the title.
Thank you for the suggestion; we made proper alterations. But, the discussion is not limited to Brazil, the essay was motivated by a discussion here, however the problem is common to other countries.
Context: open access
The manuscript discusses the assessment of open access articles by comparing Elsevier and MDPI. This seems to suggest that APC based publishing of articles is the only way to publish in open access. Given the role of SciELO in Brazil, I would at least expect a short explanation why the discussion only focuses on the two publishers.
Thank you for your observation, but the discussion could be too extensive if we decide to discuss all aspects surrounding the topic. For example, why should we publish our manuscript in regular journals? What is the aim? What is the difference between arXiv and Research Square? And it keeps going…… But, as mentioned in the manuscript, we focused on the reference [1], which motivates a discussion about MDPI, Hindawi, and Frontiers. As an opinion manuscript, we focused our discussion around the topic, elements, and arguments that give us support for our point of view.
Context: assessment
There is an extensive discussion on the merit (or lack thereof) of the Journal Impact Factor and assessment of academic s in general. While a complete discussion of this subject lies beyond the scope of this manuscript - the discussion would probably barely fit in a book - some mention of this discussion would help to put the discussed subject in context.
Our manuscript introduces an average value of the journal´s impact factor from two publishers only as an example to discuss that the quality of both can be almost the same; then, we cannot assume that if a publisher works only with APC, the quality will become worse. But it is an open argument in the text and is out of the scope of a more detailed discussion.
Conclusion
In my view, the main argument of the manuscript centres around a generational difference in opinion on the value of articles published in Elsevier vs. MDPI in a Brazilian context. I would suggest to make this more clear, both in the title and the abstract.
Thank you for the observation. We changed the title as suggested, but we could find a better way to present the text since the main idea was to discuss the wrong assumptions made by our colleagues about predatory journals and try to elucidate about the motivations around it.
Reviewer 3 Report
Comments and Suggestions for Authors
This work and approach are intriguing; however, it falls short in terms of the methodological rigor required to substantiate the assertions put forth. While your proposal shows promise and utility, it necessitates more robust argumentation and defense. Specifically, a dedicated methodology section is imperative, elucidating the procedures employed for data extraction. Additionally, a comprehensive results section, featuring not only the obtained graph but also other supplementary visuals that reinforce the presented statements, is essential. The conclusion should seamlessly tie back to the results presented. Regrettably, the validity of the claims made would be highly questionable without these crucial elements.
Author Response
Dear Reviewer, Thank you for the suggestions.
The current manuscript is not a regular Article but an OPINION Manuscript. An opinion manuscript is a document that presents the author's viewpoint or perspective on a particular topic, issue, or subject matter. It typically reflects the author's thoughts, beliefs, and arguments based on their expertise, research, or personal experiences. Opinion manuscripts can take various forms, including essays, articles, editorials, commentaries, or opinion pieces.
In an opinion manuscript, the author aims to persuade, inform, or provoke thought among readers by presenting their viewpoint clearly, concisely, and compellingly. While opinion manuscripts are often subjective, they may incorporate evidence, reasoning, and logical analysis to support the author's position.
Finally, we do hope for questions and feedback from readers; our purpose here is only to stimulate discussion. The elements inserted in our discussion are essential to support our analysis, but of course they can be, and must be better analyzed.
Round 2
Reviewer 2 Report
Comments and Suggestions for Authors
While I still recommend to make it more clear that the article is written from a Brazilian context, I think it should be published as an opinion piece.
Author Response
Thank you for the tip, we change the title to brazilian scenario
Reviewer 3 Report
Comments and Suggestions for Authors
Thanks for the clarification; it had not been identified as an opinion article.
Author Response
thank you for the revision